# Association of trabecular bone score and bone mineral apparent density with the severity of bone fragility in children and adolescents with osteogenesis imperfecta: A cross-sectional study

Yasuhisa Ohata[1] , Taichi Kitaoka[1] , Takeshi Ishimi[1], Chieko Yamada[1], Yukako Nakano[1], Kenichi Yamamoto[1,2¤], Shinji Takeyari[1], Hirofumi Nakayama[1,3], Makoto Fujiwara[1], Takuo Kubota[1], Keiichi Ozono[1] *

1 Department of Pediatrics, Osaka University Graduate School of Medicine, Suita, Osaka, Japan,
2 Department of Statistical Genetics, Osaka University Graduate School of Medicine, Suita, Osaka, Japan,
3 The 1st. Department of Oral and Maxillofacial Surgery, Osaka University Graduate School of Dentistry, Suita, Osaka, Japan

☯ These authors contributed equally to this work.
¤ Current address: Department of Children's and Women's Health, Division of Health Sciences, Osaka University Graduate School of Medicine, Suita, Osaka, Japan
* keioz@ped.med.osaka-u.ac.jp

## Abstract

Osteogenesis imperfecta (OI) is a hereditary skeletal disease characterized by bone fragility. Areal bone mineral density (BMD), evaluated by dual-energy X-ray absorptiometry (DXA), is used to assess bone brittleness. The height-adjusted BMD Z-score ($BMD_{HAZ}$) is calculated in children and adolescents with OI to reduce the confounding factor of short stature. However, even with the $BMD_{HAZ}$, severity evaluation in children and adolescents with OI is challenging because certain abnormalities in bone quality cannot be accurately assessed by BMD analysis. The trabecular bone scores (TBS) and bone mineral apparent density (BMAD), which represent the structural integrity of bone and bone-size-associated BMD, respectively, are associated with fracture risk. Recently, age- and sex-specific reference ranges have been reported, enabling the calculation of Z-scores for children. To evaluate which density measurements show the highest correlation with fracture risk, we analyzed the associations between the Z-scores of TBS, BMAD, and $BMD_{HAZ}$, fracture rate, and genetic variants. We retrospectively reviewed 42 participants with OI aged 5 to 20 years who underwent DXA. *COL1A1/2* pathogenic variants were detected in 41 of the 42 participants. In participants with nonsense and frameshift variants (n = 17) resulting in haploinsufficiency and mild phenotype, the TBS Z-score was negatively correlated with fracture rate (FR) (r = -0.50, *p* = 0.042). In participants with glycine substitution (n = 9) causing the severe phenotype, the BMAD Z-scores were negatively correlated with FR (r = -0.74, *p* = 0.022). No correlation between the $BMD_{HAZ}$ and FR was observed in both groups. These findings suggest that the TBS and BMAD are useful in assessing children and adolescents with OI with specific genetic variants.

**Data Availability Statement:** All relevant data are within the paper and its Supporting Information files (S1 Table).

**Funding:** This study was supported by "the Japan Agency for Medical Research and Development: https://www.amed.go.jp/en/" (No. 22ek0109549h0002, 22bm0804006h0206 and J210705007) to K.O., "the Ministry of Health, Labor, and Welfare: https://www.mhlw.go.jp/english/" (No. 22FC1012) to T.Ku., and "the Japan Society for the Promotion of Science: https://www.jsps.go.jp/english/" (No. 21H02881) to K.O.. The funders had no role in study design, data collection and analysis, decision to publish, or preparation of the manuscript.

**Competing interests:** The authors have declared that no competing interests exist.

## Introduction

Osteogenesis imperfecta (OI) is a rare inheritable skeletal disorder, caused, in most cases, by structural or quantitative defects in the α1 and α2 chains of collagen type I, encoded by *COL1A1* or *COL1A2* genes, respectively [1, 2]. These patients show bone fragility, deformities of long bones, and short stature. The bone brittleness results in fractures even after minor trauma and subsequent growth restriction [3]. The clinical classification of OI was first established by Sillence and colleagues in 1979 [4] and five clinical forms are defined in the revised "Nosology and Classification of Genetic Skeletal Disorders." [5] OI type I has the mildest phenotype while type IV is a moderately severe form. Among the non-lethal forms of OI, type III patients are most severely affected. It has been reported that a quantitative deficiency in collagen type I causes a mild form of OI [6], while a structural defect derived from glycine substitutions in *COL1A1* or *COL1A2* genes results in a more severe form [7].

It is very important to evaluate the severity of bone fragility in children and adolescents with OI to make a diagnosis and provide appropriate treatment and lifestyle management advice. Evaluation of areal bone mineral density (BMD) using dual-energy x-ray absorptiometry (DXA) can aid OI diagnosis [8–10], given that OI patients have significantly lower BMD compared to healthy individuals of the same age and sex [11, 12]. However, bone density measurement in growing bone is challenging due to the inherent complexities of measuring the projection density of objects of varying sizes. Moreover, the thickness of the bone itself affects the projected density. Bone density is strongly related to height; thus, short stature may also contribute to low BMD [13]. As a result, The International Society for Clinical Densitometry has recommended a height-adjustment approach for DXA measurements of lumbar spine in short stature children [14] using the height-for-age Z-score (HAZ)-adjusted spine areal BMD-for-age Z-score ($BMD_{HAZ}$) [13]. Bone mineral apparent density (BMAD) calculation, in which bone area is transformed to bone volume of each vertebra to estimate the effects of bone depth and bone size, is also endorsed for clinical evaluation of skeletal features [14, 15]. Previously, two studies evaluated BMD in children and adolescents with OI using different parameters. Rauch et al. studied the relationship between genotype and lumbar spinal areal BMD Z-score with adjustment for age, sex, and height Z-scores [16], while, Diacinti et al. examined the mean BMAD before and after intravenous neridronate therapy [17]. Although neither study assessed bone fragility, BMD Z-scores, and BMAD may be useful approaches to evaluate bone fragility in children and adolescents with OI.

Other skeletal features, including bone microarchitecture are also associated with bone strength and risk of fracture. These aspects of the bone are known as bone quality [18], which can be assessed by bone biopsies and high-resolution peripheral quantitative computed tomography (HR-pQCT) [19, 20]. Indeed, bone quality is impaired not only in animal models of OI [21, 22] but also in OI human bones [23–25]. Furthermore, microscopic mechanical properties are lower in bones of children with OI [26] and negatively correlate with disease severity [27, 28]. To estimate the bone fragility, it is necessary to perform a combined evaluation of bone strength (measured by bone density) and bone quality (reflected by the bone structure and material properties), especially in bones with underlying molecular defects that are associated with altered material properties, such as in patients with OI. While the aforementioned findings provide the rationale for performing bone biopsy routinely in OI assessment of children and adolescents, this approach is difficult to implement given the invasiveness of bone biopsy and the specialized expertise required for its widespread use. Furthermore, HR-pQCT has much higher costs and more limited accessibility than that of DXA [29–31].

Trabecular bone score (TBS) is a gray level texture index extracted from a DXA image of the lumbar spine which evaluates pixel gray level variation and serves as an indirect index of

trabecular architecture [32]. While TBS is an indirect analysis of microarchitecture in trabecular bone, it is supposed to estimate fracture risk based on the correlation with connectivity density, trabecular number, and trabecular separation [33–35]. Actually, it has been reported that TBS is lower in severe adult OI patients [36]. Although with a small sample size, Rehberg et al. analyzed TBS in children with OI and suggested TBS to be a useful tool for monitoring the skeletal changes [37]. These findings possibly imply that TBS can complement standard density measures in assessing skeletal integrity in children and adolescents with OI. Nevertheless, TBS Z-score was not previously available for children and adolescents due to the lack of reference data. Recently, however, reliable age- and sex-specific reference ranges of TBS and BMAD have been reported [38, 39], enabling the calculation of the Z-score for individuals aged from 5 to 20 years. Fraga et al. evaluated the Z-score of TBS for healthy Brazilian children and adolescents [40]. Additionally, the BMAD Z-score has been used for cancer survivors [41] and arthrogryposis patients [42] to assess bone health conditions in children and adolescent populations. However, the relationship between these scores and bone fragility were not evaluated in these studies, and no publication has yet assessed these Z-scores in children and adolescents with OI. To evaluate the clinical applicability of TBS and BMAD as an assessment of bone fragility in children and adolescents with OI, we analyzed the associations between the Z-scores of TBS, BMAD, and $BMD_{HAZ}$, fracture rate, and genetic variants in this cross-sectional study.

## Materials and methods

### Study population

This cross-sectional study was approved by the Institutional Review Board (IRB) of Osaka University (IRB number 688, 15601, and 19535). Written informed consent was obtained from participants aged 16 years or older. For participants under 16 years of age, parental written informed consent was obtained. Additionally, consent from individuals over 8 years of age was also obtained in written form. All procedures for this study followed the ethical standards of the institution and the 1964 Helsinki declaration. Between 2015 to 2022, a total of 44 participants who had received a genetic diagnosis of OI underwent DXA at Osaka University, ranging in age from 5 to 20 years. Among them, two participants were excluded from enrollment: one due to the unavailability of a written informed consent form, and another because of the inability to conduct an accurate DXA scan following lumbar spine surgery for scoliosis correction. Thus, overall, 42 participants with OI harboring pathogenic variants were included in this study. Participants with lumbar fractures were excluded from the study since they did not undergo DXA. If DXA was performed several times in an individual participant during this observational period, we selected the latest data and analyzed them (N = 39). One of the authors (T.Ki., T.Ku., or K.O.) assessed each participant and assigned classifications according to the Sillence classification. All individuals reported here were Asian ethnicity resided in Japan.

### Genetic analysis

As we previously reported [43], we performed targeted next-generation sequencing (NGS) and whole exome sequencing (WES). All of the candidate pathogenic variants detected by targeted NGS and WES were confirmed by Sanger sequencing using a 3730 DNA analyzer (Thermo Fisher Scientific, Waltham, MA, USA). If genetic analysis had not been performed, but a pathogenic variant was identified in an affected family member, the detected variant was considered to be the causative variant for the participant.

### Anthropometric measurements

When the participants underwent DXA, their body weight (kg) and height (cm) were measured at our hospital using a digital electronic scale and stadiometer, respectively, and the body mass index (BMI) was calculated. If the participants could not stand correctly, we measured their recumbent height using an inelastic tape. HAZ and Z-scores of body weight and BMI were calculated based on reference data from The Japanese Society for Pediatric Endocrinology [44].

### Fracture assessment

Each participant or their parents reported their clinical history of bone fractures at every DXA scan, including vertebral and non-vertebral fractures confirmed radiographically. To verify this information, we also reviewed the participants' medical charts and counted the number of fractures retrospectively. To calculate the annual fracture rate, we divided the number of fractures up to the DXA measurement by the age at the DXA scan.

### DXA assessment

Areal BMD ($g/cm^2$) at the lumbar spine (L1-L4) of all participants was measured by a Hologic Discovery A DXA scanner (from 2015 to 2020) and a Hologic Horizon A DXA scanner (from 2021 to 2022). We further confirmed cross-calibration between these DXA machines. DXA data were analyzed using version 13.5 software (Hologic Inc., Bedford, MA, USA). All measurements were analyzed for bone mineral content, bone area, and aBMD at the department of radiology of our hospital by a professional technician. $BMD_{HAZ}$ was evaluated using the data from previous studies [45–47]. Spine BMAD was calculated as previously reported [39], Z-score was determined using the reference and LMS values [39]. To evaluate TBS, the existing data were collected and analyzed by TBS iNsight software (ver 3.03, Medimaps, Plan-les Ouates, Switzerland). The Z-score of TBS was calculated using the reference and LMS values by sex and age [38].

### Statistical analysis

All statistical analyses were performed with JMP® Pro software version 15.0.0 (SAS institute Inc., Cary, NC, USA). The Shapiro-Wilk test was used to determine the distribution of continuous data. Normally distributed variables were expressed as mean ± standard deviation (SD). Non-normally distributed variables were expressed as median (interquartile range [IQR]). To determine the association between two variables, Pearson correlation was used for normal distribution data, while Spearman rank correlation was used for non-normally distributed data. The Wilcoxon signed-rank test was used to compare the difference in non-normally distributed continuous variables between the two groups. The difference with $p$ value $< 0.05$ was considered statistically significant.

## Results

### Characteristics of study participants

In this study, 42 participants with a genetically confirmed clinical diagnosis of OI from 40 unrelated families were enrolled. A total of 22 males and 20 females were included. The median age of the participants at the DXA measurement was 13.6 years (range: 4.99–20.4). The distribution of the Sillence classification was as follows: type I (n = 31); type III (n = 5); type IV (n = 5); type V (n = 1). Genetic analysis was performed all participants and the distribution of detected variants among 42 individuals was as follows: *COL1A1* (n = 32), *COL1A2*

**Table 1. Characteristics of the study population.**

| | OI participants |
|---|---|
| Sex, Male/Female | 22/20 |
| Age at DXA (years) | 13.6 [10.4, 18.9] (4.99, 20.4) |
| Silence classification | I, 31; III, 5; IV, 5; V, 1 |
| Genetic analysis | *COL1A1*, 32 |
| | *COL1A2*, 9 |
| | *IFITM5*, 1 |
| Tx at DXA measurement (tx, number of participants) | no treatment, 14 |
| | risedronate, 7 |
| | pamidronate, 9 |
| | alendronate, 4 |
| | zoledronic acid, 4 |
| | eldecalcitol, 4 |

Data are presented as mean ± SD or median [interquartile range] and (range). DXA, dual-energy x-ray absorptiometry; OI, osteogenesis imperfecta; Tx, treatment.

(n = 9), and *IFITM5* (n = 1) genes. At the time of DXA analysis, 14 participants had no treatment for OI, while 24 received bisphosphonate (7 treated with risedronate, 9 treated with pamidronate, 4 treated with alendronate, and 4 treated with zoledronic acid) and 4 received active vitamin D analogue (eldecalcitol) (Table 1). Among 14 individuals without any treatment at the DXA scan, 13 had been treated previously (5 received risedronate and pamidronate, 5 received pamidronate and alendronate, 1 received pamidronate, 1 received alendronate, and 1 received alendronate and active vitamin D analogue [alfacalcidol]). Detailed data of the study participants are presented in S1 Table.

## Assessment of bone fragility by Z-scores of $BMD_{HAZ}$, BMAD, and TBS

The annual fracture rate was considered as an index of bone fragility. We analyzed the relationship of the fracture rate with the Z-scores of $BMD_{HAZ}$, BMAD, and TBS to determine which factors are associated with OI fracture risk. When we analyzed all participants, there was no correlation between the fracture rate and Z-scores of $BMD_{HAZ}$ ($\rho$ = -0.0031, $p$ = 0.98), BMAD ($\rho$ = -0.093, $p$ = 0.56), and TBS ($\rho$ = -0.19, $p$ = 0.22) (Fig 1).

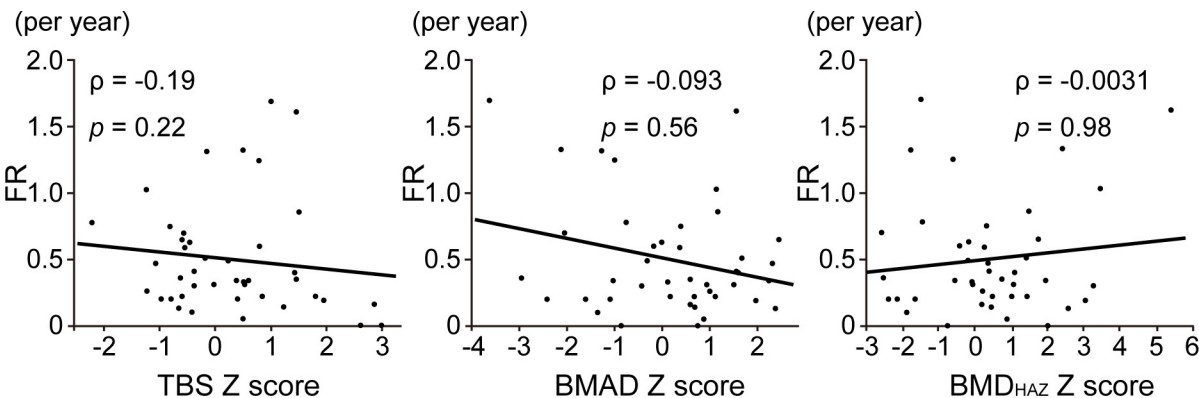

**Fig 1. Association of bone fragility with DXA parameters in all OI participants.** Correlation between annual fracture rate (FR) and Z-scores of TBS, BMAD, and $BMD_{HAZ}$ in OI participants (n = 42) by Spearman rank.

**Table 2. Characteristics of the study population with haploinsufficiency and glycine substitution variant in *COL1A1* and *COL1A2* genes.**

| | Haploinsufficiency | glycine substitution |
|---|---|---|
| Sex, Male/Female | 9/8 | 4/5 |
| Age at DXA (years) | 13.8 ± 5.23 | 11.5 ± 3.77 |
| Sillence classification | I, 17 | I, 2; III, 3; IV, 4 |
| Genetic analysis | *COL1A1*, 17 | *COL1A1*, 4<br>*COL1A2*, 5 |
| Height Z-score | -0.82 ± 0.89 | -3.63 ± 2.26 |
| Body weight Z-score | -0.49 ± 1.37 | -2.34 ± 2.24 |
| BMI Z-score | -0.04 ± 1.12 | -0.08 ± 0.92 |
| Annual fracture rate (incidence/year) | 0.33 ± 0.20 | 0.74 ± 0.50 |
| $BMD_{HAZ}$ Z-score | 0.13 ± 1.34 | 1.27 ± 1.85 |
| BMAD Z-score | 0.21 ± 1.37 | 0.12 ± 1.83 |
| TBS Z-score | 0.27 ± 1.10 | 0.28 ± 1.02 |

Data are presented as mean ± SD. DXA, dual-energy x-ray absorptiometry; BMI, body mass index, $BMD_{HAZ}$ Z-score, bone mineral density-for-age Z-score adjusted for height-for-age Z-score; BMAD, bone mineral apparent density; TBS, trabecular bone score.

Forty one participants had pathogenic variants in *COL1A1* or *COL1A2* genes. Nonsense and frameshift variants in one *COL1A1* allele result in haploinsufficiency and mild phenotype. On the other hand, the glycine substitution variants, either in *COL1A1* or *COL1A2* genes, cause severe OI [6, 7, 43]. In this study, 17 individuals had nonsense (n = 6) and frameshift (n = 11) variants in *COL1A1* gene while 9 had glycine substitution variant either in *COL1A1* (n = 4) or *COL1A2* (n = 5) genes (Table 2). To evaluate non-severe subset of OI, we analyzed only individuals with haploinsufficient variants (n = 17) and found that their TBS Z-score was negatively correlated with annual fracture rate (r = -0.50, *p* = 0.042) (Fig 2). In addition, after excluding participants with Sillence type III to extract non-severe participants, the TBS Z-score was still negatively correlated with annual fracture rate (r = -0.38, *p* = 0.022, S1 Fig). On the other hand, in glycine substitution group, annual fracture rate was negatively correlated with the Z-score of BMAD (r = -0.74, *p* = 0.022) (Fig 3).

## Discussion

For the first time in existing literature, our study determined the correlations between Z-scores of TBS, BMAD, and $BMD_{HAZ}$ and fracture risk in children and adolescents with OI.

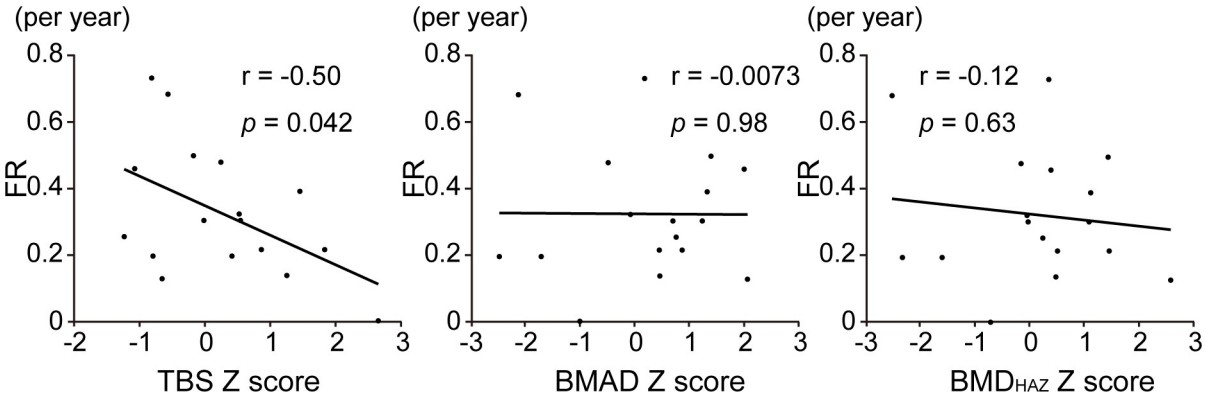

**Fig 2. Association of bone fragility with DXA parameters in haploinsufficient defect group.** Correlation between annual fracture rate (FR) and Z-scores of TBS, BMAD, and $BMD_{HAZ}$ in OI participants with *COL1A1* haploinsufficient variants (n = 17) by Pearson correlation.

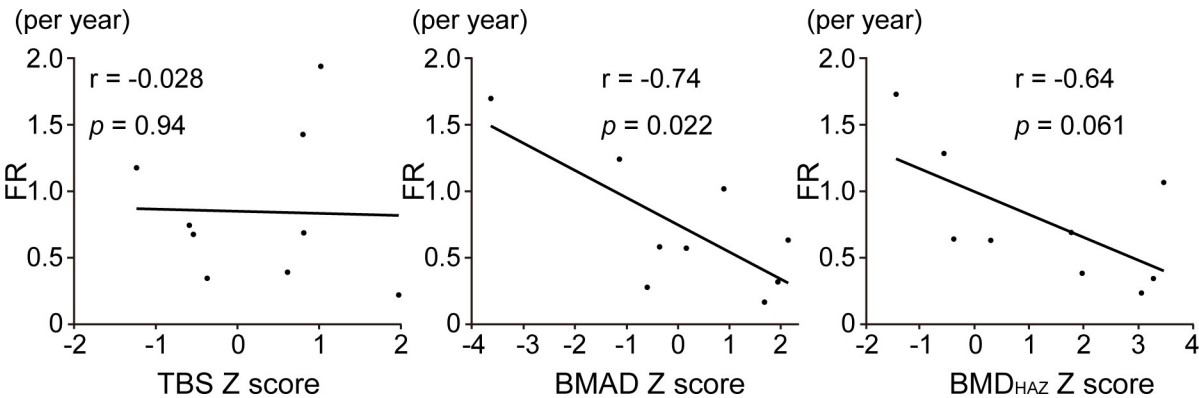

**Fig 3. Association of bone fragility with DXA parameters in glycine substitution group.** Correlation between annual fracture rate (FR) and Z-scores of TBS, BMAD, and BMD$_{HAZ}$ in OI participants with glycine substitution variants in *COL1A1* or *COL1A2* (n = 9) by Pearson correlation.

Interestingly, the TBS Z-score was associated with the fracture rate only in genetically stratified non-severe OI participants. When we extracted clinically non-severe subset by excluding individuals with Sillence type III, the TBS Z-score was also negatively correlated with annual fracture rate. (S1 Fig). Through pQCT evaluation, it was observed that areas of high and low BMD were interspersed within the same bone in children and adolescents with mild OI [23]. Although BMD by a phantom-based measurement is almost uniformly low, direct assessments of bone using quantitative backscattered electron imaging or ash of bone revealed the presence of patchy mineral increases in OI bone [1]. TBS is estimated by analyzing the variogram of the projected image of the region of interest in which the sum of the squares of the difference in gray-level between pixels at the determined distance are calculated [29]. From these fundamental mechanisms, it is postulated that TBS can assess the trabecular microarchitecture. In fact, several *ex vivo* studies showed correlations between TBS and connectivity density, trabecular number, and trabecular separation [33–35]. If a "patchy" abnormality of mineralization in mild OI bones, which contributes to the bone fragility, is detectable by TBS, our result would be of clinical significance. However, the mechanism by which the haploinsufficient variants contribute to the observed abnormalities in OI bones needs to be elucidated. Furthermore, it is essential to clarify whether the dosage of bisphosphonate can impact the correlation between TBS and fracture rate, as patients with more severe bone fragility often receive intensified treatment regimens. Further investigations with HR-pQCT or pQCT may provide valuable insights into the underlying pathophysiology of OI.

Although the mechanism by which the glycine substitution in *COL1A1* and *COL1A2* genes result in short stature remained incompletely understood [48], we and other groups have reported that OI patients with glycine substitution are distinguishable not only by bone fragility but also by short stature [43, 49]. Consistent with previous reports, individuals with glycine substitution had a greater annual fracture rate and were significantly shorter than those with haploinsufficient variants in this study (S2 Fig). The BMAD and BMD$_{HAZ}$ are calculated by considering bone size and short stature, respectively. Our findings revealed that BMAD correlated with bone fragility only in severe OI, suggesting that an abnormality of bone size, which is corrected by BMAD measurements, can cause inaccuracies in the classical assessment of BMD for these participants. Conversely, when the BMAD is applied to adjust for the smallness of bones, the measurement of BMD by DXA can effectively evaluate bone fragility including bone quality in severe cases of OI. Although the Z-score of BMAD and BMD$_{HAZ}$ were correlated with each other in this study in all participants as well as in the glycine substitution group

(S3 Fig), only BMAD Z-score correlated with fracture rate. This finding may suggest that the formula used in BMAD calculation is more appropriate to correct for the bone size effect in severe OI of children and adolescents. Furthermore, it suggests that considering a measurement of BMAD that accounts for bone size leads to a stronger correlation with fracture rates.

This study has some limitations in its design. Although we calculated the Z-scores of TBS and BMAD based on the reference of non-African Americans [38, 39], there was no information on how many Asian participants were included in the reference data. Second, the TBS reference data published by Kalkwarf et al. was based on TBS iNsight software pre-release version 4.0. In this study, we analyzed the TBS by version 3.03 as we were unable to use the pre-release model; additionally, we could not evaluate the difference between these different models. Third, Rehberg et al reported that there was no significant difference of TBS between before and after bisphosphonate treatment in children with OI ($p = 0.25$) [37]. However, ideally, this study should have been carried out in bisphosphonate-naïve patients, as it is well-known that bisphosphonate can positively affect the BMD in the bones of patients with OI [50]. Unfortunately, we could not evaluate the effects of any OI treatments, including bisphosphonate, because of the retrospective cross-sectional nature of the study, and many participants had already received OI treatments at the time of the investigation (S1 Table). Therefore, in the future, it will be necessary to conduct a longitudinal study to analyze the impact of treatments on our findings. Fourth, the sample size was small, particularly in the subgroup analysis, because of the rarity of OI. However, we confirmed that the subgroup used in our study was statistically appropriate through statistical review and that our study has the potential to serve as a pilot study for future, larger, multi-center studies. Finally, we analyzed the fracture rate as a marker of severity of bone fragility in participants with OI aged 5 to 20 years. The risk of fractures can vary depending on the age of patients, as patient activity level, growth rate, and the stage of puberty are completely different between early childhood and adolescence. We need to analyze more patients with OI in a narrow age range to rule out such bias.

In summary, the current study revealed that the Z-score of TBS can evaluate the bone fragility in mild children and adolescents with OI, while that of BMAD can assess bone fragility in severe OI. These findings suggested the calculation of Z-scores of TBS and BMAD is useful for the assessment of children and adolescents with OI. Furthermore, appropriate assessment method must be selected based on the genetic variants of OI patients.

## Supporting information

**S1 Table. Detailed data of the study participants.** Age, age at the DXA scan; FR, the number of fractures up to the DXA scan divided by the age; $BMD_{HAZ}$, height-for-age Z-score-adjusted bone mineral density-for-age Z-score; BMAD, Z-score of bone mineral apparent density; TBS, Z-score of trabecular bone score; Ht-SD, standard deviation score of height; BW-SD, standard deviation score of body weight; Tx, treatment at the DXA scan; past Tx, past treatment history; RIS, risedronate, PAM, pamidronate; ALN, alendronate; ZOL, zoledronic acid; Elde, eldecalcitol; Alfa, alfacalcidol; MSCT, mesenchymal stem cell transplantation in utero; none, no treatment; n.d., not detected; † We previously confirmed this deletion variant causing exon 21 skipping (p.Gly364_Arg399del) by mRNA analysis (Takeyari S, Kubota T, Ohata Y, Fujiwara M, Kitaoka T, Taga Y, et al. 4-Phenylbutyric acid enhances the mineralization of osteogenesis imperfecta iPSC-derived osteoblasts. J Biol Chem. 2021;296:100027. Epub 20201123. doi: 10. 1074/jbc.RA120.014709. PubMed PMID: 33154166; PubMed Central PMCID: PMC7948972.). (DOCX)

**S1 Fig. Correlation between annual fracture rate (FR) and Z-scores of TBS, BMAD, and BMD$_{HAZ}$ in non-severe OI participants without individuals with Sillence type III (n = 37).** (TIF)

**S2 Fig. Comparison of the annual fracture rate and height Z-score (Ht-SD) between OI participants harboring nonsense and frameshift variants in *COL1A1* causing haploinsufficient defect (HI, n = 17) and glycine substitution (GS) either in *COL1A1* or *COL1A2* causing severe phenotype (n = 9).** (TIF)

**S3 Fig. Correlation between Z-scores of BMAD and BMD$_{HAZ}$ in all participants (n = 42) and in individuals with glycine substitution either in *COL1A1* or *COL1A2* (n = 9).** (TIF)

## Acknowledgments

We would like to thank Ms. Satomi Okamura, Mr. Kazutaka Nishio, and Mr. Eisuke Hida for assistance with the statistical review. We would also like to thank Editage (www.editage.com) for English language editing. Finally, we thank the study participants for consenting to participate in this study.

## Author Contributions

**Conceptualization:** Yasuhisa Ohata, Taichi Kitaoka, Takuo Kubota, Keiichi Ozono.

**Data curation:** Yasuhisa Ohata, Taichi Kitaoka, Yukako Nakano, Shinji Takeyari, Makoto Fujiwara.

**Formal analysis:** Yasuhisa Ohata, Taichi Kitaoka.

**Funding acquisition:** Takuo Kubota, Keiichi Ozono.

**Investigation:** Yasuhisa Ohata, Taichi Kitaoka, Takeshi Ishimi, Chieko Yamada, Kenichi Yamamoto, Shinji Takeyari.

**Methodology:** Yasuhisa Ohata, Taichi Kitaoka, Hirofumi Nakayama, Makoto Fujiwara.

**Project administration:** Taichi Kitaoka.

**Resources:** Taichi Kitaoka.

**Supervision:** Keiichi Ozono.

**Validation:** Yasuhisa Ohata, Taichi Kitaoka.

**Visualization:** Yasuhisa Ohata.

**Writing – original draft:** Yasuhisa Ohata.

**Writing – review & editing:** Taichi Kitaoka, Makoto Fujiwara, Takuo Kubota, Keiichi Ozono.

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
