## [Decision Letter · Decision Letter 0]

3 May 2023

PONE-D-23-03588Association of trabecular bone score and bone mineral apparent density with the severity of bone fragility in children and adolescents with osteogenesis imperfecta: A cross-sectional studyPLOS ONE

Dear Dr. Ozono,

Thank you for submitting your manuscript to PLOS ONE. After careful consideration, we feel that it has merit but does not fully meet PLOS ONE’s publication criteria as it currently stands. Therefore, we invite you to submit a revised version of the manuscript that addresses the points raised during the review process.

 In line with PLOS ONE publication criteria and as noted by both reviewers, please ensure that the statistical analysis is described in sufficient detail so that another investigator could reproduce the results. Please also more clearly state the study objectives. Additional comments are listed below. 

We look forward to receiving your revised manuscript.

Kind regards,

Heather Macdonald, Ph.D

Academic Editor

PLOS ONE

Journal Requirements:

Additional Editor Comments:

As noted by Reviewer 1, assessing bone outcomes in children and adolescents with osteogenesis imperfecta is challenging for a number of reasons, and while the authors are to be commended for taking on this challenge, there are a number of limitations that need to be addressed more thoroughly. In particular, the authors are encouraged to more clearly state their study objectives, and ensure that the steps in their statistical analysis align appropriately with these objectives. Statistical review may be warranted, particularly given the small sample size (please clarify in the methods whether this was a convenience sample, or whether a sample size calculation was performed a priori). I am also unclear why there are more supplementary tables and figures than are included in the main paper.

Additional comments

1. Subjects, participants and patients are used interchangeably throughout the methods/results/figures - suggest using participants throughout.

2. Abstract: Consider modifying to include actual numerical results in the abstract.

3. Lines 58-61: Please clarify that DXA measures two-dimensional areal BMD, which as Reviewer 1 highlights is strongly influenced by body size.

4. Line 61: Quotes are not needed around the name.

5. Line 66: Change "are also" to "is also"

6. Lines 67-68: The authors state that these approaches may be useful for evaluation of bone fragility in children and youth with OI. If they haven't been used in a previous study of children/youth with OI, please clarify this, or provide the appropriate references if they have been used previously.

7. Line 79: The radiation dose is comparable between HR-pQCT and DXA - please update the text accordingly (see papers such as Pezzuti et al., JPEM, https://doi.org/10.1515/jpem-2016-0252).

8. Line 105: Regarding the cohort of 47 OI participants, were these all of the OI patients who underwent DXA at the Osaka University Hospital, or do the 47 represent the proportion who consented to participate in study? If it is the latter, please clarify how many individuals were invited to participate.

9. Line 107: How many participants had multiple DXA scans between 2015 and 2022? Depending on the number of participants with repeat scans, have the authors explored any longitudinal analyses?

10. Line 113: How many participants were excluded due to an inaccurate DXA scan?

11. Lines 115-6: Change to All individuals in the cohort were of Asian ethnicity and resided in Japan. Related to this, how many children of Asian ethnicity were included in the non-African American reference dataset published by Kawlkwarf et al?

12. Lines 125-6: I assume height and weight were measured at the time of the DXA scan? Please clarify in the text, and mention that BMI was calculated.

13. Line 127: The authors mention that HAZ were calculated using reference data from the JSPE (quotes not needed), but weight-for-age and BMI-for-age Z scores are provided in Table 1- were they calculated using the same reference data? Please add this information.

14. Line 131: Did participants report their fracture history at the time of the DXA scan? And for participants who underwent repeat DXA scans during 2015-2022, did they report their fracture history at each DXA scan? And earlier in the methods, the authors mentioned that the OI diagnosis was based on low trauma fractures, so it would help to clarify if fracture history only included low/minimal trauma fractures, or low, moderate and high trauma fractures. It would also help to report the total number of fractures in the cohort.

15. Line 137: Consider starting the sentence with Areal BMD (instead of the abbreviation), and provide the units for aBMD. Did the same technician acquire and analyze all DXA scans? How was cross-calibration confirmed? What is the %CV for the DXA outcomes and TBS outcomes? In addition, the TBS reference data published by Kawlkwarf et al. was based on TBS iNsight software pre-release version 4.0, not version 3.03 as used by the authors in the present study - please discuss how that impacts this analysis.

16. Statistics: As noted by both Reviewers, additional details are required to ensure that the description of the statistical analysis aligns with the study objectives. Please clarify how regression assumptions were checked.

17. Table 1: Change gender to sex unless the authors specifically asked about gender/socially constructed roles instead of biological sex.

18. Line 188-190: Modify wording since "effect" implies causation, which is not appropriate in this cross-sectional study.

19. Lines 230-231: I don't follow the wording of this sentence.

20. Line 243: Change to: needs to be elucidated.

21. Was the study adequately powered to conduct the subgroup analyses?  

Reviewers' comments:

Reviewer's Responses to Questions

**Comments to the Author**

1. Is the manuscript technically sound, and do the data support the conclusions?

Reviewer #1: Yes

Reviewer #2: Partly

2. Has the statistical analysis been performed appropriately and rigorously? 

Reviewer #1: Yes

Reviewer #2: I Don't Know

3. Have the authors made all data underlying the findings in their manuscript fully available?

Reviewer #1: No

Reviewer #2: Yes

4. Is the manuscript presented in an intelligible fashion and written in standard English?

Reviewer #1: Yes

Reviewer #2: Yes

5. Review Comments to the Author

Reviewer #1: Association of trabecular bone score and bone mineral apparent density with the severity of bone fragility in children and adolescents with osteogenesis imperfecta

This study examines the relationship between fracture rates and different DXA measures of bone density in OI adolescents.

General:

This study tackles a very challenging topic. Bone density in growing bone is challenging because of the limitations of projection measures when size is changing. On top of that, adding bones that do not grow as they should, bones that are pathologic, and bones that have been treated by various bisphosphonates for various amounts of time, you have a very challenging area of study. The authors are applauded for finding any correlations in the data set at all! It would help to clarify the challenges in the introduction. Be explicit that predicting fracture rates must combine estimates of strength (density) and quality (structure and material properties). Explain why projection measures of density of different size objects is challenging. Explain that DXA is a common tool for assessing osteoporosis risk in elderly but poses challenges when applied to children of all different sizes, especially when the defect is a molecular defect which likely alters material properties, not just density (how much bone) and structure (where the bone is). This will help make it stand out how challenging it is. The first paragraph of the intro just hints at “confounding factor of short stature”. Explain it more.

The underlying problem in a projection measurement is the thickness of the bone affects the projected density. It is therefore not surprising that the measure that adjusts for the thickness of the bone (BMAD) best correlates. Other measures of adjusting (age-adjusted, height-adjusted, height-for-age Zscore) are using alternative measures of size to try to do this.

What is the clinical question? Severity of fracture risk? (line 30) “severity evaluation in children and adolescents is hard because of abnormalities in bone quality that BMD cannot accurately assess”. It is true, bone quality cannot be assessed with a density measure. Significant research has shown density correlates well with strength but not with “quality”, which generally means everything but strength. Especially in OI, the underlying problem is the material itself so no matter how much bone they have and how it is arranged at the macro level, there will be a material integrity problem. Fracture risks is not directly quality, however. It is a combination of strength (density) and quality (arrangement, composition, material properties). It seems like what you are asking is: can we predict fracture risk with a density measure of some sort? You should bring this out in the abstract. It is currently phrased “to evaluate the clinical applicability [of adjusted density measures] we analyzed associations.” Be more direct: we determined correlations between different DXA measures of bone density and fracture risk.

Explain the rationale for analyzing particular genetic variants separately. Would you expect specific variants to be more correlated with density measures than others? Why? Justify this briefly in the intro.

Abstract:

Focus the abstract to the question: which density measures best correlates with fracture risk?

Intro:

Line 59: “BMD is lower in OI bones” Is this because the bones are smaller? Beware!

Line 95: “To evaluate clinical applicability” It seems like you are saying: “finally data sets are available for bone density in children that adjust for size; let’s see if they work for pathologic bone in which density is secondary to a material defect.” Is that the goal? Have these data sets (Z scores) been used for other non-pathologic bone density problems (eating disorders, amenorrhea, etc.) in which the bone tissue itself is fine there is just very little of it. Do the scores predict fracture rates in these populations (who are granted much less prone to fracture)?

Line 78: What is meant by “bone biospies are too professional to be performed widely”?

Line 89: “TBS can be performed to evaluate bone quality in OI children” Perhaps instead “TBS can complement standard density measures in assessing skeletal integrity”

Line 170: How they diagnosed with OI if they do not have a genetic variant? Is it purely a symptomatic diagnosis?

Line 188: Change “affected” to “correlated”. Are these slist

Line 190: “tended to show” This is overly generous for a correlation of 0.27 and a p value of 0.07. Even if it were significant, it is still a poor correlation.

Supplementary Figure 2 is important. Can you bring it to main text? Then you could leave out stats from the paragraph and it will read much easier. Make the BMAD correlation bold in the table as that is the only one that is significant (and negative).

Line 190: Why do you think BMADHaz had a positive effect? Isn’t that worrying? Wat is the issue with this measure do you think? Discuss in discussion.

Line 192: “We analyzed the relationship of fracture rate with Z-scores.” How is this different from line 186 “regression analysis for fracture rate with Z-scores”. You need to make it clear that line 186 is the multiple regression analysis and line 192 is the simple linear regression. What does it mean that BMAD has a strong association (multiple regression) but a weak correlation (simple linear regression)?

Line 225: “usefulness”? Why do you single out BMADHaz? Just say “For the first time we determined correlations between density measures Z-scores of TBS, BMAD, and BMADHaz with fracture risk.”

Line 243: How would you elucidate these? With HRpqCT or pqCT to determine structural deficiencies?

Line 252: “abnormality of bone size have a strong impact on BMD” This makes it sound like having small bones means you have low BMD. This is an artefact of the imaging technique and using a projection method. This should be made more clear.

Line 253: “BMD itself” It it is adjusted for size, then it isn’t BMD, it is BMAD, right?

Discussion: Why do you think that for haploinsufficiency fractures were not correlated with BMAD but glycine fractures were? Whereas haploinsufficiency fractures were correlated with TBS but glycine fractures were not?

Reviewer #2: This study evaluates the recently described Z-scores for several parameters (TBS and BMAD) calculated from retrospective DEXA results in children and adolescents with OI. It compares fracture rates with these measures and the more standard DEXA parameter: height and age-adjusted BMD Z-score. It also analyses the correlation between the Z-scores and fracture rates in mild and severe OI subgroups. There was a negative correlation of fracture rate with TBS in the patients with mild OI and BMAD in patients with severe OI.

Abstract and Introduction:

1. Line 27: the way this is phrased is confusing “The height-for-age Z-score (HAZ)-adjusted BMD-for-age Z-score (BMDHAZ)”. I would suggest simplifying it to something like: “the height-adjusted BMD Z-score for age (BMDHAZ)” or the “height-for-age BMD Z-score (BMDHAZ)”

2. Line 78: suggest changing the word “professional” to “specialized”

3. Line 96: suggest changing “bone fragility” to “fracture rate”

Results:

1. Table 1: this is a little difficult to read. Suggest perhaps breaking up the OI participants into the subgroups you use for the later analysis. Having the number and demographics (average heights, fracture rates and DEXA parameters etc) of the patients with haploinsufficiency vs glycine mutations in this table would be useful.

2. Although you explain your reasoning for performing both the multiple regression and the simple linear regression (lines 153-155), the conflicting results are confusing. The multiple regression found the BMAD Z-score was significantly negatively correlated, and the BMD-HAZ was significantly positively correlated with fracture rates. However, the simple linear regression found no correlation for either of these DEXA parameters (in the whole cohort). A statistician’s input may be helpful here to determine the most appropriate test to present.

Figures:

1. I don’t think Figure 1 is required in the main paper, as it is only to prove why you are using these genetic results to represent your “mild” vs “severe” OI phenotype groups. You can simply state the result in the text and include the figure with Figure S3.

2. Figure 2 and 3: suggest use “per year” instead of “/yr.”

3. I suggest Figure S1 be included in the main paper (it shows no correlation when the entire cohort is analysed using simple linear regression) as Fig 2 and 3 (the “mild/haploinsufficiency” and “severe/glycine mutation” subgroups) should be interpreted with this in mind.

Discussion:

1. Lines 228-229: A result is listed in the discussion that is not mentioned earlier in the paper. Suggest adding this to the results section.

2. The postulated explanation for why TBS only correlates in mild OI due to haploinsufficiency is interesting. It would be interesting to know if this correlation simply relates to the amount of bisphosphonate treatment the patient has received (with more severe cases often receiving more treatment) – this is partially addressed later in the discussion

3. Line 251: suggest changing “evaluated the” to “correlated with”

4. Line 257-259: it would be useful to consider that a measure (BMAD) that accounts for the size of the bones results in a stronger correlation with fracture rates due to the mechanical properties of the bone (rather than just providing a better estimate of density). Small/narrow bones break more easily than large bones, even if they have the same density. BMAD might reflect this.

5. Line 269: missing words “we analysed fracture rate as a (marker of) severity”

6. Line 270-273: the difference fracture rates with age are not just due to activity levels. Growth rate and puberty also play a role.

General comments:

1. Suggest using the wording “children and adolescents with OI” rather than “OI children and adolescents” throughout (e.g. line 86 and 89)

6. PLOS authors have the option to publish the peer review history of their article (what does this mean?). If published, this will include your full peer review and any attached files.

Reviewer #1: No

Reviewer #2: No

---

## [Author Response · Author response to Decision Letter 0]

5 Jul 2023

RESPONSE TO REVIEWERS

Additional Editor Comments:

As noted by Reviewer 1, assessing bone outcomes in children and adolescents with osteogenesis imperfecta is challenging for a number of reasons, and while the authors are to be commended for taking on this challenge, there are a number of limitations that need to be addressed more thoroughly. In particular, the authors are encouraged to more clearly state their study objectives, and ensure that the steps in their statistical analysis align appropriately with these objectives. Statistical review may be warranted, particularly given the small sample size (please clarify in the methods whether this was a convenience sample, or whether a sample size calculation was performed a priori). I am also unclear why there are more supplementary tables and figures than are included in the main paper.

Response: Thank you for your thorough review and important comments. We agree with the comments from the Editor and Reviewer 1 that it is challenging to assess growing bone with molecular defects. According to the suggestion of Reviewer 1, we have added sentences to clarify the difficulty of this study as follows. “However, bone density measurement in growing bone is challenging due to the inherent complexities of measuring the projection density of objects of varying sizes. Moreover, the thickness of the bone itself affects the projected density.”

We also agree with the comment that when predicting fracture rates, we must combine estimates of the strength (density) and quality (structure and material properties) of bones, especially in pathologic bones. To further explain this, we have added the following description in the introduction section, “To estimate the bone fragility, it is necessary to perform a combined evaluation of bone strength (measured by bone density) and bone quality (reflected by the bone structure and material properties), especially in bones with underlying molecular defects that are associated with altered material properties, such as in patients with OI.” 

To clarify the study objective and clinical questions as Reviewer 1 pointed out, we have revised the description regarding clinical questions in the abstract as follows. “To evaluate which density measurements show the highest correlation with fracture risk, we analyzed the associations between the Z-scores of TBS, BMAD, and BMDHAZ, fracture rate, and genetic variants.” 

As the Editor and Reviewer 2 recommended, the manuscript underwent a statistical review by statisticians. They pointed out that performing multiple regression analysis with the whole cohort can be misleading and should be deleted from our analysis because some parameters are distributed non-normally. Thus, we have excluded the multiple regression analysis and related descriptions. We also discussed with the statisticians about the small sample number, especially in the subgroup analysis. To address the concerns regarding sampling, we have clarified the inclusion criteria and included only participants harboring any pathogenic variants. Finally, we have changed the original Supplemental Figure 1 to Figure 1 and created Table 2 as Reviewer 2 suggested. Thanks to the thorough review and constructive opinions, we believe that the manuscript has been improved and refined. 

Additional comments

1. Subjects, participants and patients are used interchangeably throughout the methods/results/figures - suggest using participants throughout.

Response: Thank you for your suggestion. We have corrected “subjects” and “patients” to “participants” throughout the methods, results, and figures.

2. Abstract: Consider modifying to include actual numerical results in the abstract.

Response: We have included the numerical results in the abstract as follows. In lines 46–48, we have added (r = -0.50, p = 0.042) and (r = -0.74, p = 0.022), respectively.

3. Lines 58-61: Please clarify that DXA measures two-dimensional areal BMD, which as Reviewer 1 highlights is strongly influenced by body size.

Response: Thank you for your suggestion. As Reviewer 1 suggested, we have added sentences as follows to explain that bone density evaluation in growing bone is challenging. “However, bone density measurement in growing bone is challenging due to the inherent complexities of measuring the projection density of objects of varying sizes. Moreover, the thickness of the bone itself affects the projected density.”

4. Line 61: Quotes are not needed around the name.

Response: As per your comment, we have excluded the quotes from the description of “The International Society for Clinical Densitometry”.

5. Line 66: Change "are also" to "is also"

Response: We have changed “are also” to “is also” in line 80 , which is highlighted with red font.

6. Lines 67-68: The authors state that these approaches may be useful for evaluation of bone fragility in children and youth with OI. If they haven't been used in a previous study of children/youth with OI, please clarify this, or provide the appropriate references if they have been used previously.

Response: Thank you for your suggestion. Rauch et al. analyzed the relationship between genotype and lumbar spinal areal BMD Z-score with adjustment for age, sex, and height Z-scores. (JBMR 2010: 1367) Diacinti et al. evaluated the mean bone mineral apparent density (BMAD) before and after intravenous neridronate therapy in children and adolescents with OI (Bone. 2021: 115608). However, they did not assess the fracture rate, as we have evaluated. We have added the following sentences to the revised manuscript with the appropriate references. “Previously, two studies evaluated children and adolescents with OI using different parameters. Rauch et al. studied the relationship between genotype and lumbar spinal areal BMD Z-score with adjustment for age, sex, and height Z-scores, while, Diacinti et al. examined the mean BMAD before and after intravenous neridronate therapy. Although both studies did not assess bone fragility, these approaches may be useful for its evaluation in children and adolescents with OI.”

7. Line 79: The radiation dose is comparable between HR-pQCT and DXA - please update the text accordingly (see papers such as Pezzuti et al., JPEM, https://doi.org/10.1515/jpem-2016-0252).

Response: Thank you for enlightening us on this matter. Based on your comments, we have deleted the description stating that HRpQCT employs a greater radiation dose than DXA.

8. Line 105: Regarding the cohort of 47 OI participants, were these all of the OI patients who underwent DXA at the Osaka University Hospital, or do the 47 represent the proportion who consented to participate in study? If it is the latter, please clarify how many individuals were invited to participate.

Response: Thank you for your suggestion. As Reviewer 1 suggested, we have included the participants who were diagnosed with OI using genetic testing. We have described this information in the revised manuscript as follows. “Between 2015 to 2022, a total of 44 participants who had received a genetic diagnosis of OI underwent DXA at Osaka University, ranging in age from 5 to 20 years. Among them, two participants were excluded from enrollment: one due to the unavailability of a written informed consent form, and another because of the inability to conduct an accurate DXA scan following lumbar spine surgery for scoliosis correction. Thus, overall, 42 participants with OI harboring pathogenic variants were included in this study. Participants with lumbar fractures were excluded from the study since they did not undergo DXA.”

9. Line 107: How many participants had multiple DXA scans between 2015 and 2022? Depending on the number of participants with repeat scans, have the authors explored any longitudinal analyses?

Response: Among 42 participants, 39 had multiple DXA scans during the study period. We have added this information to the revised manuscript as follows. “If DXA was performed several times in an individual participant during this observational period, we selected the latest data and analyzed them (N = 39).” Unfortunately, we did not perform longitudinal analyses because this is a cross-sectional study. We are interested in performing such analyses and had described it as a study limitation as follows. “Therefore, in the future, it will be necessary to conduct a longitudinal study to analyze the impact of treatments on our findings.”

10. Line 113: How many participants were excluded due to an inaccurate DXA scan?

Response: Thank you for your comment. One participant was excluded because of the inability to conduct an accurate DXA scan following lumbar spine surgery for scoliosis correction. We have included this information in the revised manuscript (kindly see response to comment 8).

11. Lines 115-6: Change to All individuals in the cohort were of Asian ethnicity and resided in Japan. Related to this, how many children of Asian ethnicity were included in the non-African American reference dataset published by Kawlkwarf et al?

Response: Thank you for your comment. Unfortunately, Kalkwarf et al. categorized the participants as having African ancestry or non-African ancestry based on parental report, and there was no detailed description on whether participants of Asian ethnicity were included in the non-African group. Kindler et al. reported that ancestry was categorized as black or non-black, the latter of which include people of European, Hispanic, Asian, and other ancestries. However, there was no description about the precise number of Asian participants. We have added the following sentences as a limitation of this study. “Although we calculated the Z-scores of TBS and BMAD based on the reference of non-African Americans, there was no information on how many Asian participants were included in the reference data.”

12. Lines 125-6: I assume height and weight were measured at the time of the DXA scan? Please clarify in the text, and mention that BMI was calculated.

Response: Thank you for your kind comment. As you assumed, we measured height and weight at the time of the DXA scan. We have corrected the sentence as follows to clearly state that BMI was calculated from these measurements, “When the participants underwent the DXA scan, their body weight (kg) and height (cm) were measured at our hospital using a digital electronic scale and stadiometer, respectively, and the body mass index (BMI) was calculated.”

13. Line 127: The authors mention that HAZ were calculated using reference data from the JSPE (quotes not needed), but weight-for-age and BMI-for-age Z scores are provided in Table 1- were they calculated using the same reference data? Please add this information.

Response: As you pointed out, we calculated weight-for-age and BMI-for-age Z scores from the same reference data from the JSPE. We have changed the sentence as follows. “HAZ and Z-scores of body weight and BMI were calculated based on reference data from The Japanese Society for Pediatric Endocrinology”. In the revised description, we have excluded the quotes.

14. Line 131: Did participants report their fracture history at the time of the DXA scan? And for participants who underwent repeat DXA scans during 2015-2022, did they report their fracture history at each DXA scan? And earlier in the methods, the authors mentioned that the OI diagnosis was based on low trauma fractures, so it would help to clarify if fracture history only included low/minimal trauma fractures, or low, moderate and high trauma fractures. It would also help to report the total number of fractures in the cohort.

Response: Thank you for your thorough review. As you commented, the participants reported their fracture history at every DXA scan between 2015-2022. We have added the phrase “at every DXA scan” in the fracture assessment. Among the 42 participants, we had actually included participants who had been diagnosed clinically based on a history of fractures with mild trauma. However, as Reviewer 1 suggested, we have included only the participants who had been diagnosed with OI through genetic testing. In the revised manuscript, we have removed the descriptions regarding the clinical diagnosis of OI. The total number of fractures in each participant was included in S1 Table.

15. Line 137: Consider starting the sentence with Areal BMD (instead of the abbreviation), and provide the units for aBMD. Did the same technician acquire and analyze all DXA scans? How was cross-calibration confirmed? What is the %CV for the DXA outcomes and TBS outcomes? In addition, the TBS reference data published by Kawlkwarf et al. was based on TBS iNsight software pre-release version 4.0, not version 3.03 as used by the authors in the present study - please discuss how that impacts this analysis.

Response: Thank you for your suggestion and comment. We have changed the start of the paragraph from aBMD to Areal BMD and added the unit (g/cm2). A single, professional technician performed and analyzed the DXA scans, and he has confirmed that the data measured by the Hologic Discovery A DXA scanner and Hologic Horizon A DXA scanner are identical. We have revised the description as follows, “All measurements were analyzed for bone mineral content, bone area, and aBMD at the department of radiology of our hospital by a professional technician.” 

Unfortunately, we could not get any data regarding the difference between TBS iNsight version 4.0 and 3.03 because the version 4.0 software is a pre-release model. We have added the following description as a study limitation in the discussion section. “Second, the TBS reference data published by Kawlkwarf et al. was based on TBS iNsight software pre-release version 4.0. In this study, we analyzed the TBS by version 3.03 as we were unable to use the pre-release model; additionally, we could not evaluate the difference between these different models.”

16. Statistics: As noted by both Reviewers, additional details are required to ensure that the description of the statistical analysis aligns with the study objectives. Please clarify how regression assumptions were checked.

Response: As the Editor and Reviewer 2 recommended, the manuscript underwent a statistical review by statisticians. They pointed out that performing multiple regression analysis with the whole cohort can be misleading and should be deleted from our analysis because some parameters are distributed non-normally. Thus, we have excluded the multiple regression analysis and related descriptions.

17. Table 1: Change gender to sex unless the authors specifically asked about gender/socially constructed roles instead of biological sex.

Response: We have changed “gender” to “sex” in Table 1.

18. Line 188-190: Modify wording since "effect" implies causation, which is not appropriate in this cross-sectional study.

Response: Thank you for your thorough review. Following the statistical review, this sentence has been deleted (kindly see response to comment 16). 

19. Lines 230-231: I don't follow the wording of this sentence.

Response: We apologize for the confusing description. We wished to explain that using pQCT, Rauch et al. reported there are high and low BMD areas interspersed within the same bone in children and adolescents with mild OI. To refine it, we have changed the sentence as follows, “Through pQCT evaluation, it was observed that areas of high and low BMD were interspersed within the same bone in children and adolescents with mild OI.”

20. Line 243: Change to: needs to be elucidated.

Response: Thank you for your correction. We have changed the sentence as follows, “However, the mechanism by which the haploinsufficient variants contribute to the observed abnormalities in OI bones need to be elucidated.” 

21. Was the study adequately powered to conduct the subgroup analyses?

Response: Thank you for your comment. We have discussed this point with the statisticians during the statistical review. They concluded that we cannot exclude the possibility that the power was not enough, especially in the subgroup analysis as you said, due to the lack of calculation of the optimal sample number prior to conducting this study. Although they provided this feedback on our statistical analysis, they also acknowledged that the subgroup used in our study was appropriate since we had established clear eligibility criteria and enrolled as many participants as possible. They also recognized that our study has the potential to serve as a pilot study for future, larger studies with multiple institutions. We have added the following sentences as a study limitation in the discussion section. “However, we confirmed that the subgroup used in our study was statistically appropriate through statistical review and that our study has the potential to serve as a pilot study for future, larger, multi-center studies.”

Reviewer #1: Association of trabecular bone score and bone mineral apparent density with the severity of bone fragility in children and adolescents with osteogenesis imperfecta

This study examines the relationship between fracture rates and different DXA measures of bone density in OI adolescents.

General:

This study tackles a very challenging topic. Bone density in growing bone is challenging because of the limitations of projection measures when size is changing. On top of that, adding bones that do not grow as they should, bones that are pathologic, and bones that have been treated by various bisphosphonates for various amounts of time, you have a very challenging area of study. The authors are applauded for finding any correlations in the data set at all! It would help to clarify the challenges in the introduction. Be explicit that predicting fracture rates must combine estimates of strength (density) and quality (structure and material properties). Explain why projection measures of density of different size objects is challenging. Explain that DXA is a common tool for assessing osteoporosis risk in elderly but poses challenges when applied to children of all different sizes, especially when the defect is a molecular defect which likely alters material properties, not just density (how much bone) and structure (where the bone is). This will help make it stand out how challenging it is. The first paragraph of the intro just hints at "confounding factor of short stature". Explain it more.

The underlying problem in a projection measurement is the thickness of the bone affects the projected density. It is therefore not surprising that the measure that adjusts for the thickness of the bone (BMAD) best correlates. Other measures of adjusting (age-adjusted, height-adjusted, height-for-age Z score) are using alternative measures of size to try to do this.

What is the clinical question? Severity of fracture risk? (line 30) "severity evaluation in children and adolescents is hard because of abnormalities in bone quality that BMD cannot accurately assess". It is true, bone quality cannot be assessed with a density measure. Significant research has shown density correlates well with strength but not with "quality", which generally means everything but strength. Especially in OI, the underlying problem is the material itself so no matter how much bone they have and how it is arranged at the macro level, there will be a material integrity problem. Fracture risks is not directly quality, however. It is a combination of strength (density) and quality (arrangement, composition, material properties). It seems like what you are asking is: can we predict fracture risk with a density measure of some sort? You should bring this out in the abstract. It is currently phrased "to evaluate the clinical applicability [of adjusted density measures] we analyzed associations." Be more direct: we determined correlations between different DXA measures of bone density and fracture risk.

Response: Thank you for your thorough review and insightful suggestions. As you have suggested, we have added sentences as follows to explain that the evaluation of the bone density in growing bone is challenging, “However, bone density measurement in growing bone is challenging due to the inherent complexities of measuring the projection density of objects of varying sizes. Moreover, the thickness of the bone itself affects the projected density.”

We also agree with the comment that when predicting fracture rates, we must combine estimates of the strength (density) and quality (structure and material properties) of bones, especially in pathologic bones. To further explain this, we have added the following descriptions in the introduction section, “To estimate the bone fragility, it is necessary to perform a combined evaluation of bone strength (measured by bone density) and bone quality (reflected by the bone structure and material properties), especially in bones with underlying molecular defects that are associated with altered material properties, such as in patients with OI.” 

To clarify the study objective and clinical questions as Reviewer 1 pointed out, we have revised the description regarding clinical questions in the abstract as follows. “To evaluate which density measurements show the highest correlation with fracture risk, we analyzed the associations between the Z-scores of TBS, BMAD, and BMDHAZ, fracture rate, and genetic variants.” 

Explain the rationale for analyzing particular genetic variants separately. Would you expect specific variants to be more correlated with density measures than others? Why? Justify this briefly in the intro.

Response: We apologize for the lack of an explanation. We have added the following description in the introduction section. “It has been reported that a quantitative deficiency in collagen type I causes a mild form of OI, while a structural defect derived from glycine substitutions in COL1A1 or COL1A2 genes results in a more severe form.”

Abstract:

Focus the abstract to the question: which density measures best correlates with fracture risk?

Response: Thank you for your kind suggestion. We have corrected the sentence as follows, “To evaluate which density measurements show the highest correlation with fracture risk, we analyzed the associations between the Z-scores of TBS, BMAD, and BMDHAZ, fracture rate, and genetic variants.”

Intro:

Line 59: "BMD is lower in OI bones" Is this because the bones are smaller? Beware!

Response: Thank you for your thorough review. We corrected the sentence as follows to avoid being misleading. ““Evaluation of the bone mineral density (BMD) using dual-energy x-ray absorptiometry (DXA) can aid OI diagnosis, given that OI patients have significantly lower BMD levels compared to healthy individuals of the same age and sex.”

Line 95: "To evaluate clinical applicability" It seems like you are saying: "finally data sets are available for bone density in children that adjust for size; let's see if they work for pathologic bone in which density is secondary to a material defect." Is that the goal? Have these data sets (Z scores) been used for other non-pathologic bone density problems (eating disorders, amenorrhea, etc.) in which the bone tissue itself is fine there is just very little of it. Do the scores predict fracture rates in these populations (who are granted much less prone to fracture)?

Response: Thank you for your meaningful suggestion. Although Fraga et al. used the TBS Z-score, which we referred to in this manuscript, for healthy Brazilian children and adolescents, there is no publication in which the TBS Z-scores were used for other non-pathogenic bone density problems including eating disorders or amenorrhea. On the other hand, the BMAD Z-scores were used for cancer survivors (Guo et al., 2021) and arthrogryposis patients (Dahan-Oliel et al., 2020), to evaluate bone health conditions in children and adolescent populations. However, they did not analyze the relationship between the BMAD Z-scores and fracture rate. To include this background context, we have added the following descriptions with new references. “Fraga et al. evaluated the Z-score of TBS for healthy Brazilian children and adolescents. Additionally, the BMAD Z-score has been used for cancer survivors and arthrogryposis patients to assess bone health conditions in children and adolescent populations. However, the relationship between these scores and bone fragility were not evaluated in these studies, and no publication has yet assessed these Z-scores in children and adolescents with OI.” 

Line 78: What is meant by "bone biopsies are too professional to be performed widely"?

Response: We apologize for the confusing wording. Reviewer 2 has also pointed this issue out and suggested to change the word of “professional” to “specialized”. According to the suggestion, we have modified the sentence as follows. “While the aforementioned findings provide the rationale for performing bone biopsy and HRpQCT routinely in OI assessment of children and adolescents, this approach is difficult to implement given the invasiveness of bone biopsy and the specialized expertise required for its widespread use.”

Line 89: "TBS can be performed to evaluate bone quality in OI children" Perhaps instead "TBS can complement standard density measures in assessing skeletal integrity"

Response: According to your suggestion, we have altered the sentence as follows, “These findings possibly imply that TBS can complement standard density measures in assessing skeletal integrity in OI children and adolescents.” 

Line 170: How they diagnosed with OI if they do not have a genetic variant? Is it purely a symptomatic diagnosis?

Response: Thank you for your important comment. In the original manuscript, we had included some participants who have been symptomatically diagnosed with OI without confirmation of any pathogenic variants. With such inclusion criteria, some patients with juvenile osteoporosis, which is one of the most important differential diagnoses, were eligible for inclusion in this study. To refine the inclusion criteria, we have included only the participants with OI harboring pathogenic variants and re-analyzed the data in the revised manuscript. We have altered the description regarding the study population as follows. “Between 2015 to 2022, a total of 44 participants who had received a genetic diagnosis of OI underwent DXA at Osaka University, ranging in age from 5 to 20 years. Among them, two participants were excluded from enrollment: one due to the unavailability of a written informed consent form, and another because of the inability to conduct an accurate DXA scan following lumbar spine surgery for scoliosis correction. Thus, overall, 42 participants with OI harboring pathogenic variants were included in this study. 

Line 188: Change "affected" to "correlated". 

Response: Thank you for your correction. Reviewer 2 raised concerns about the propriety of the multiple regression analysis and proposed that statisticians conduct a statistical review to ascertain the suitability of this analysis. The statisticians concluded that multiple regression analysis with the whole cohort can be misleading and should be deleted from our analysis because some parameters are distributed non-normally. Accordingly, we have removed the multiple regression analysis and its relevant descriptions, including the original line 188.

Line 190: "tended to show" This is overly generous for a correlation of 0.27 and a p value of 0.07. Even if it were significant, it is still a poor correlation.

Response: As mentioned above, we have deleted the description regarding the multiple regression analysis, including the original line 190.

Supplementary Figure 2 is important. Can you bring it to main text? Then you could leave out stats from the paragraph and it will read much easier. Make the BMAD correlation bold in the table as that is the only one that is significant (and negative).

Response: Thank you for your kind suggestion. Reviewer 2 also suggested the result of the original supplementary Figure 2 should appear in the main text. We have added the following sentence just after the description concerning Figure 2. “In addition, after excluding participants with Sillence type Ⅲ to extract non-severe participants, the TBS Z-score was still negatively correlated with annual fracture rate (r = -0.38, p = 0.022, S1 Fig).” The original Supplementary Figure 2 has been re-named Supplementary Figure 1 because the original Supplementary Figure 1 has been changed to Figure 1 as Reviewer 2 had instructed.

Line 190: Why do you think BMADHaz had a positive effect? Isn't that worrying? Wat is the issue with this measure do you think? Discuss in discussion.

Response: We apologize for the confusing results of the multiple regression analysis. Reviewer 2 also raised concerns about this issue and had proposed that statisticians conduct a statistical review to ascertain the suitability of this analysis. The statisticians concluded that multiple regression analysis with the whole cohort can be misleading and should be deleted from our analysis because some parameters are distributed non-normally. Accordingly, we have removed the multiple regression analysis and its relevant descriptions.

Line 192: "We analyzed the relationship of fracture rate with Z-scores." How is this different from line 186 "regression analysis for fracture rate with Z-scores". You need to make it clear that line 186 is the multiple regression analysis and line 192 is the simple linear regression. What does it mean that BMAD has a strong association (multiple regression) but a weak correlation (simple linear regression)?

Response: As described above, we have deleted the multiple regression analysis and its relevant descriptions, which included the sentence that states BMAD has a strong association in multiple regression but a weak correlation in simple linear regression.

Line 225: "usefulness"? Why do you single out BMADHaz? Just say "For the first time we determined correlations between density measures Z-scores of TBS, BMAD, and BMADHaz with fracture risk."

Response: Thank you for your kind suggestion. As you suggested, we have altered the description as follows. “For the first time in existing literature, our study determined the correlations between density measures Z-scores of TBS, BMAD, and BMDHAZ and fracture risk in children and adolescents with OI.”

Line 243: How would you elucidate these? With HRpqCT or pqCT to determine structural deficiencies?

Response: Thank you for your stimulating comment. As you proposed, we agree that research with HRpQCT or pQCT may be able to help elucidate the pathophysiology of OI. Reviewer 2 suggested that the amount of bisphosphonate can also have an impact on these mechanisms, and we have added a related sentence. We then added the following description to reflect your comment. “Further investigations with HRpQCT or pQCT may provide valuable insights into the underlying pathophysiology of OI.”

Line 252: "abnormality of bone size have a strong impact on BMD" This makes it sound like having small bones means you have low BMD. This is an artefact of the imaging technique and using a projection method. This should be made more clear.

Response: As you pointed out, the original description can be misleading. We have altered the description as follows. “Our findings revealed that BMAD correlated with bone fragility only in severe OI, suggesting that an abnormality of bone size, which is corrected by BMAD measurements, can cause inaccuracies in the classical assessment of BMD for these participants.”

Line 253: "BMD itself" It it is adjusted for size, then it isn't BMD, it is BMAD, right?

Response: Thank you for your thorough review. We agree that the original wording can be misleading. We have corrected the description as follows. “Conversely, when the BMAD is applied to adjust for the smallness of bones, the measurement of BMD by DXA can effectively evaluate bone fragility including bone quality in severe cases of OI.”

Discussion: Why do you think that for haploinsufficiency fractures were not correlated with BMAD but glycine fractures were? Whereas haploinsufficiency fractures were correlated with TBS but glycine fractures were not?

Response: Thank you for your important question. We originally wrote that the formula used in BMAD calculation is more appropriate to correct for the bone size effect in severe OI harboring glycine substitution. In addition, Reviewer 2 suggested that it would be useful to consider that a measurement of BMAD results in a stronger correlation with fracture rates due to the mechanical properties of the bone rather than just providing a better estimate of density. Reviewer 2 also commented that small and narrow bones can break more easily than large bones, even if they have the same density, and BMAD might reflect this. We agree with these comments and have added the following descriptions in the discussion section. “Furthermore, it suggests that considering a measurement of BMAD that accounts for bone size leads to a stronger correlation with fracture rates, likely due to the consideration of mechanical properties of the bone.”

Reviewer #2: This study evaluates the recently described Z-scores for several parameters (TBS and BMAD) calculated from retrospective DEXA results in children and adolescents with OI. It compares fracture rates with these measures and the more standard DEXA parameter: height and age-adjusted BMD Z-score. It also analyses the correlation between the Z-scores and fracture rates in mild and severe OI subgroups. There was a negative correlation of fracture rate with TBS in the patients with mild OI and BMAD in patients with severe OI.

Response: Thank you for your review and many insightful comments and suggestions. According to your review, we have modified our manuscript as follows.

Abstract and Introduction:

1. Line 27: the way this is phrased is confusing "The height-for-age Z-score (HAZ)-adjusted BMD-for-age Z-score (BMDHAZ)". I would suggest simplifying it to something like: "the height-adjusted BMD Z-score for age (BMDHAZ)" or the "height-for-age BMD Z-score (BMDHAZ)"

Response: We apologize for the confusing description. According to your suggestion, we rephrased to “the height-adjusted BMD Z-score for age (BMDHAZ)".

2. Line 78: suggest changing the word "professional" to "specialized"

Response: Thank you for your kind suggestion. Accordingly, we have changed the word of “professional” to “specialized” and the corrected sentence is “While the aforementioned findings provide the rationale for performing bone biopsy and HRpQCT routinely in OI assessment of children and adolescents, this approach is difficult to implement given the invasiveness of bone biopsy and the specialized expertise required for its widespread use.” 

3. Line 96: suggest changing "bone fragility" to "fracture rate"

Response: According to your suggestion, we have rephrased “bone fragility” to “fracture rate”.

Results:

1. Table 1: this is a little difficult to read. Suggest perhaps breaking up the OI participants into the subgroups you use for the later analysis. Having the number and demographics (average heights, fracture rates and DEXA parameters etc) of the patients with haploinsufficiency vs glycine mutations in this table would be useful.

Response: Thank you for your kind suggestion. Accordingly, we have excluded data of Z-score of height, body weight, BMI, BMDHAZ, BMAD, and TBS, and annual fracture rate from Table 1. Then, we have divided the participants into haploinsufficiency and glycine variant groups and showed the participant number and demographics as Table 2.

2. Although you explain your reasoning for performing both the multiple regression and the simple linear regression (lines 153-155), the conflicting results are confusing. The multiple regression found the BMAD Z-score was significantly negatively correlated, and the BMD-HAZ was significantly positively correlated with fracture rates. However, the simple linear regression found no correlation for either of these DEXA parameters (in the whole cohort). A statistician's input may be helpful here to determine the most appropriate test to present.

Response: Thank you so much for your accurate advice. As you and the Editor recommended, the manuscript underwent a statistical review by statisticians. They pointed out that performing multiple regression analysis with the whole cohort can be misleading and should be deleted from our analysis because some parameters are distributed non-normally. Thus, we have excluded the multiple regression analysis and related descriptions.

Figures:

1. I don't think Figure 1 is required in the main paper, as it is only to prove why you are using these genetic results to represent your "mild" vs "severe" OI phenotype groups. You can simply state the result in the text and include the figure with Figure S3.

Response: We agree with your suggestion. We have described the result regarding the original Figure 1 in the discussion section and have included it in the original Figure S3 (Figure S2 in the revised manuscript). 

2. Figure 2 and 3: suggest use "per year" instead of "/yr."

Response: We have altered the description of “ /yr.” to “per year” in all figures.

3. I suggest Figure S1 be included in the main paper (it shows no correlation when the entire cohort is analyzed using simple linear regression) as Fig 2 and 3 (the "mild/haploinsufficiency" and "severe/glycine mutation" subgroups) should be interpreted with this in mind.

Response: Thank you for your kind suggestion. Accordingly, we have included the original Figure S1 in the main paper as Figure 1.

Discussion:

1. Lines 228-229: A result is listed in the discussion that is not mentioned earlier in the paper. Suggest adding this to the results section.

Response: Thank you for your suggestion. Accordingly, we have added a description in the results section as follows: “In addition, after excluding participants with Sillence type Ⅲ to extract non-severe participants, the TBS Z-score was still negatively correlated with annual fracture rate (r = -0.38, p = 0.022, S1 Fig).

2. The postulated explanation for why TBS only correlates in mild OI due to haploinsufficiency is interesting. It would be interesting to know if this correlation simply relates to the amount of bisphosphonate treatment the patient has received (with more severe cases often receiving more treatment) – this is partially addressed later in the discussion

Response: Thank you for your thoughtful suggestion. We agree that it would be interesting to assess whether the correlation simply relates to the amount of BP treatment. We have added the following sentences in the discussion section. “Furthermore, it is essential to clarify whether the dosage of bisphosphonate can impact the correlation between TBS and fracture rate, as patients with more severe bone fragility often receive intensified treatment regimens.”

3. Line 251: suggest changing "evaluated the" to "correlated with"

Response: We have altered the description to “correlated with” in the referred site.

4. Line 257-259: it would be useful to consider that a measure (BMAD) that accounts for the size of the bones results in a stronger correlation with fracture rates due to the mechanical properties of the bone (rather than just providing a better estimate of density). Small/narrow bones break more easily than large bones, even if they have the same density. BMAD might reflect this.

Response: Thank you very much for your important insight. We agree with your suggestion and have added a description regarding this issue as follows, “Furthermore, it suggests that considering a measurement of BMAD that accounts for bone size leads to a stronger correlation with fracture rates, likely due to the consideration of mechanical properties of the bone.”

5. Line 269: missing words "we analyzed fracture rate as a (marker of) severity"

Response: Thank you for your thorough review. We have added the words “marker of” in the referred site.

6. Line 270-273: the difference fracture rates with age are not just due to activity levels. Growth rate and puberty also play a role.

Response: We agree with your suggestion and have revised the description as follows, “The risk of fractures can vary depending on the age of patients, as patient activity level, growth rate, and the stage of puberty are completely different between early childhood and adolescence.”

General comments:

1. Suggest using the wording "children and adolescents with OI" rather than "OI children and adolescents" throughout (e.g. line 86 and 89)

Response: Thank you for your suggestion. We have changed the phrase "OI children and adolescents" to "children and adolescents with OI".

---

## [Decision Letter · Decision Letter 1]

10 Aug 2023

PONE-D-23-03588R1Association of trabecular bone score and bone mineral apparent density with the severity of bone fragility in children and adolescents with osteogenesis imperfecta: A cross-sectional studyPLOS ONE

Dear Dr. Ozono,

Thank you for submitting your manuscript to PLOS ONE. After careful consideration, we feel that it has merit but does not fully meet PLOS ONE’s publication criteria as it currently stands. Therefore, we invite you to submit a revised version of the manuscript that addresses the points raised during the review process.

We look forward to receiving your revised manuscript.

Kind regards,

Heather Macdonald, Ph.D

Academic Editor

PLOS ONE

Journal Requirements:

**Additional Editor Comments:**

Thank you for addressing the reviewers' comments. Reviewer 2 has one comment on the Discussion: Page 18 end of paragraph 1: suggest removing “likely due to the consideration of mechanical properties of the bone.” BMAD accounts for bone size but doesn’t specifically account for other mechanical properties of bone.

I have a few other minor comments:

1. In the abstract, start sentence 2 with Areal bone mineral density (BMD). Similarly, on page 5, add areal to the second sentence and remove "the" (Evaluation of areal bone mineral density). Please also remove levels after "lower BMD" in this sentence. Further on in this paragraph, consider remove "of this" after "As a result" and remove the aBMD abbreviation after spine areal BMD.

2. End of pg 5/start of page 6: I find the sentence that begins with "Previously, two studies evaluated..." rather vague. Please reword (i.e., evaluated BMD?). Similarly, the last sentence of this paragraph (that begins with "Although both studies..." could be reworded - perhaps "Although neither study assessed bone fragility, BMD Z-scores and BMAD may be useful approaches to evaluate bone fragility in children and adolescents with OI" (or something like that!).

3. Please change HRpQCT to HR-pQCT throughout the manuscript.

4. Bottom of page 6: By "its widespread use" are the authors referring to HR-pQCT?

5. Discussion, first sentence: "density measures Z-scores of TBS..." could be reworded. In the 2nd sentence, change "risk of fracture" to "fracture rate". Please remove numerical results from the discussion - these should only appear in the results section.

Reviewers' comments:

Reviewer's Responses to Questions

**Comments to the Author**

1. If the authors have adequately addressed your comments raised in a previous round of review and you feel that this manuscript is now acceptable for publication, you may indicate that here to bypass the “Comments to the Author” section, enter your conflict of interest statement in the “Confidential to Editor” section, and submit your "Accept" recommendation.

Reviewer #1: All comments have been addressed

Reviewer #2: All comments have been addressed

2. Is the manuscript technically sound, and do the data support the conclusions?

Reviewer #1: Yes

Reviewer #2: Yes

3. Has the statistical analysis been performed appropriately and rigorously? 

Reviewer #1: Yes

Reviewer #2: Yes

4. Have the authors made all data underlying the findings in their manuscript fully available?

Reviewer #1: Yes

Reviewer #2: Yes

5. Is the manuscript presented in an intelligible fashion and written in standard English?

Reviewer #1: Yes

Reviewer #2: Yes

6. Review Comments to the Author

Reviewer #1: All comments have been addressed adequately. The authors are congratulated on an interesting study. Always be careful when using DEXA to assess bones of different size!

Reviewer #2: All comments have been addressed well.

One additional minor comment:

Discussion

Page 18 end of paragraph 1:

I suggest removing “likely due to the consideration of mechanical properties of the bone.”

BMAD accounts for bone size but doesn’t specifically account for other mechanical properties of bone.

7. PLOS authors have the option to publish the peer review history of their article (what does this mean?). If published, this will include your full peer review and any attached files.

Reviewer #1: No

Reviewer #2: No

---

## [Author Response · Author response to Decision Letter 1]

16 Aug 2023

RESPONSE TO REVIEWERS

Additional Editor Comments:

Thank you for addressing the reviewers' comments. Reviewer 2 has one comment on the Discussion: Page 18 end of paragraph 1: suggest removing “likely due to the consideration of mechanical properties of the bone.” BMAD accounts for bone size but doesn’t specifically account for other mechanical properties of bone.

Response: Thank you for your thorough review. As the Reviewer 2 has suggested, we have deleted the wording “likely due to the consideration of mechanical properties of the bone.” from the Discussion section.

I have a few other minor comments:

1. In the abstract, start sentence 2 with Areal bone mineral density (BMD). Similarly, on page 5, add areal to the second sentence and remove "the" (Evaluation of areal bone mineral density). Please also remove levels after "lower BMD" in this sentence. Further on in this paragraph, consider remove "of this" after "As a result" and remove the aBMD abbreviation after spine areal BMD.

Response: Per your comments, we have started the sentence 2 with Areal bone mineral density (BMD) in the abstract. On page 5, we have revised the sentence to read “Evaluation of areal bone mineral density.” We have removed the words “levels” and “of this,” as well as the abbreviation “aBMD” from the specified location. 

2. End of pg 5/start of page 6: I find the sentence that begins with "Previously, two studies evaluated..." rather vague. Please reword (i.e., evaluated BMD?). Similarly, the last sentence of this paragraph (that begins with "Although both studies..." could be reworded - perhaps "Although neither study assessed bone fragility, BMD Z-scores and BMAD may be useful approaches to evaluate bone fragility in children and adolescents with OI" (or something like that!).

Response: Thank you for your suggestion. To make it clear, we have changed the sentence to “Previously, two studies evaluated BMD in children and adolescents with OI using different parameters.” We also have changed the last sentence of this paragraph as you suggested “Although neither study assessed bone fragility, BMD Z-scores, and BMAD may be useful approaches to evaluate bone fragility in children and adolescents with OI.”

3. Please change HRpQCT to HR-pQCT throughout the manuscript.

Response: We have changed “HRpQCT” to “HR-pQCT” on page 6 and 17.

4. Bottom of page 6: By "its widespread use" are the authors referring to HR-pQCT?

Response: Thank you for your thorough check. We have intended for the term “widespread use” to specifically refer to bone biopsy. To make it clear, we deleted HR-pQCT from this sentence and reworded it as follows, “While the aforementioned findings provide the rationale for performing bone biopsy routinely in OI assessment of children and adolescents, this approach is difficult to implement given the invasiveness of bone biopsy and the specialized expertise required for its widespread use.”

5. Discussion, first sentence: “density measures Z-scores of TBS…” could be reworded. In the 2nd sentence, change “risk of fracture” to “fracture rate”. Please remove numerical results from the discussion – these should only appear in the results section.

Response: Thank you for your advice. Following your suggestion, we have made two changes: firstly, the sentence now reads, “For the first time in existing literature, our study determined the correlations between Z-scores of TBS, BMAD, and BMDHAZ and fracture risk in children and adolescents with OI.” Secondly, we have replaced “risk of fracture” with “fracture rate.” We have deleted the description of numerical results from the discussion section.

Reviewers' comments:

Reviewer #1: All comments have been addressed adequately. The authors are congratulated on an interesting study. Always be careful when using DEXA to assess bones of different size!

Response: Your careful and professional review greatly enhanced our manuscript. We deeply appreciate your invaluable advice and guidance.

Reviewer #2: All comments have been addressed well.

One additional minor comment:

Discussion

Page 18 end of paragraph 1:

I suggest removing “likely due to the consideration of mechanical properties of the bone.”

BMAD accounts for bone size but doesn’t specifically account for other mechanical properties of bone.

Response: Thank you for your suggestion. We have deleted the wording “likely due to the consideration of mechanical properties of the bone.” from the Discussion section. Your expert review has dramatically improved our manuscript. We sincerely appreciate your invaluable guidance.

---

## [Editor Report · Decision Letter 2]

17 Aug 2023

Association of trabecular bone score and bone mineral apparent density with the severity of bone fragility in children and adolescents with osteogenesis imperfecta: A cross-sectional study

PONE-D-23-03588R2

Dear Dr. Ozono,

We’re pleased to inform you that your manuscript has been judged scientifically suitable for publication and will be formally accepted for publication once it meets all outstanding technical requirements.

Kind regards,

Heather Macdonald, Ph.D

Academic Editor

PLOS ONE

Additional Editor Comments (optional):

Thank you for addressing the 2nd round of comments.
---

## [Editor Report · Acceptance letter]

22 Aug 2023

PONE-D-23-03588R2 

Association of trabecular bone score and bone mineral apparent density with the severity of bone fragility in children and adolescents with osteogenesis imperfecta: A cross-sectional study 

Dear Dr. Ozono:

I'm pleased to inform you that your manuscript has been deemed suitable for publication in PLOS ONE. Congratulations! Your manuscript is now with our production department. 

Kind regards, 

on behalf of

Dr. Heather Macdonald 

Academic Editor

PLOS ONE